# Improvement in Infection Prevention and Control Compliance at the Three Tertiary Hospitals of Sierra Leone following an Operational Research Study

**DOI:** 10.3390/tropicalmed8070378

**Published:** 2023-07-24

**Authors:** Rugiatu Z. Kamara, Ibrahim Franklyn Kamara, Francis Moses, Joseph Sam Kanu, Christiana Kallon, Mustapha Kabba, Daphne B. Moffett, Bobson Derrick Fofanah, Senesie Margao, Matilda N. Kamara, Matilda Mattu Moiwo, Satta S. T. K. Kpagoi, Hannock M. Tweya, Ajay M. V. Kumar, Robert F. Terry

**Affiliations:** 1United States Centers for Disease Control and Prevention Country Office, Emergency Operation Centre, Wilkinson Road, Freetown 00232, Sierra Leone; zzc0@cdc.gov; 2World Health Organization Country Office, 21 A-B Riverside Drive, Off Kingharman Road, Freetown 00232, Sierra Leone; ibrahimfkamara@outlook.com (I.F.K.); derrickfbob@gmail.com (B.D.F.); 3Ministry of Health and Sanitation, Fourth Floor, Youyi Building, Freetown 00232, Sierra Leone; franqoline@gmail.com (F.M.); samjokanu@yahoo.com (J.S.K.); christy.conteh@yahoo.com (C.K.); kabbamustapha@yahoo.co.uk (M.K.); smargao3@gmail.com (S.M.); kamaramatilda9198@gmail.com (M.N.K.); spllenz54321@gmail.com (S.S.T.K.K.); 4College of Medicine and Allied Health Science, University of Sierra Leone, Freetown 00232, Sierra Leone; 5Ministry of Defence, Republic of Sierra Leone Armed Forces, Freetown 00232, Sierra Leone; mmmoiwo@gmail.com; 6Department of Global Health, University of Washington, Seattle, WA 98195, USA; hmwtwea@yahoo.co.uk; 7International Training and Education for Health, Lilongwe P.O. Box 30369, Malawi; 8International Union against Tuberculosis and Lung Disease, 2 Rue Jean Lantier, 75001 Paris, France; akumar@theunion.org; 9International Union against Tuberculosis and Lung Disease, South-East Asia Office, C-6, Qutub Institutional Area, New Delhi 110016, India; 10Department of Community Medicine, Yenepoya Medical College, Yenepoya (Deemed to Be University), University Road, Deralakatte 75018, India; 11The Special Program for Research and Training in Tropical Diseases (TDR), World Health Organization, Avenue Appia 20, 1211 Geneva, Switzerland; terryr@who.int

**Keywords:** infection prevention and control, IPCAF, operational research, impact assessment, tertiary hospitals, antimicrobial resistance, Sierra Leone

## Abstract

Implementing infection prevention and control (IPC) programmes in line with the World Health Organization’s (WHO) eight core components has been challenging in Sierra Leone. In 2021, a baseline study found that IPC compliance in three tertiary hospitals was sub-optimal. We aimed to measure the change in IPC compliance and describe recommended actions at these hospitals in 2023. This was a ‘before and after’ observational study using two routine cross-sectional assessments of IPC compliance using the WHO IPC Assessment Framework tool. IPC compliance was graded as inadequate (0–200), basic (201–400), intermediate (401–600), and advanced (601–800). The overall compliance scores for each hospital showed an improvement from ‘Basic’ in 2021 to ‘Intermediate’ in 2023, with a percentage increase in scores of 16.9%, 18.7%, and 26.9% in these hospitals. There was improved compliance in all core components, with the majority in the ‘Intermediate’ level for each hospital IPC programme. Recommended actions including the training of healthcare workers and revision of IPC guidelines were undertaken, but a dedicated IPC budget and healthcare-associated infection surveillance remained as gaps in 2023. Operational research is valuable in monitoring and improving IPC programme implementation. To reach the ‘Advanced’ level, these hospitals should establish a dedicated IPC budget and develop long-term implementation plans.

## 1. Introduction

Infection Prevention and Control (IPC) is a cost-effective and key strategy to combat antimicrobial resistance (AMR), as it can be implemented in all sectors and settings, including those with limited resources [1]. For every infection prevented, there is one potential antibiotic treatment avoided, subsequently reducing the risk of AMR [2]. IPC is also crucial in preventing healthcare-associated infections (HAIs), another major global health challenge associated with increased mortality, morbidity, economic burden, and increased AMR [3,4].

Recognising the importance of IPC from the lessons learnt during the 2014–2016 Ebola outbreak, the Ministry of Health and Sanitation in Sierra Leone established a national IPC Programme [5]. The programme is led by the National IPC Unit (NIPCU) with a mandate to provide leadership and coordinate and monitor IPC implementation to strengthen IPC standards in all hospitals [2]. Each hospital established an IPC programme consisting of eight core components addressing different IPC aspects, in line with the WHO Guidelines on Core Components of IPC at the facility level [6]. However, data on IPC programme implementation in hospitals in Sierra Leone are scarce.

To analyse and improve IPC programme implementation at the hospital level, the WHO recommends use of the Infection Prevention and Control Assessment Framework (IPCAF) tool [7]. This is a standardised questionnaire with an associated scoring and grading system ranging from ‘inadequate’ to ‘advanced’ and an interpretation of the four-level grades (Table 1) used to assess the eight core components of an IPC programme [8]. This tool has been used by many countries to assess IPC programme implementation in hospitals.

A WHO global survey on IPC implementation in hospitals from 81 countries showed a median score denoting an advanced level of implementation, but scores were significantly lower in low-income countries [9]. Similar studies conducted in Lira Hospital in Uganda, 11 tertiary hospitals in Bangladesh, and 12 tertiary hospitals in Pakistan documented weak implementation of the hospital IPC programmes as the majority of the hospitals scored ‘basic’ according to the IPCAF grading system [8,10,11]. The use of the IPCAF has enabled the identification of several bottlenecks in implementation, including limited financial resources and competing priorities. The findings of these studies, from comparable low–middle-income countries (LMICs), align with the few studies conducted in Sierra Leone [2,12].

The most recent study on assessing IPC programme implementation in Sierra Leone using the IPCAF tool was an operational research study conducted in 2021 by Kamara et al., a co-investigator on this paper (hereafter referred to as the baseline study) [13]. The baseline study found that the IPC programmes in three tertiary hospitals (Connaught, Ola During Children’s and Princess Christian Maternity Hospitals) were sub-optimal as they scored ‘Basic’ according to the IPCAF grading. Several gaps were identified in the eight core components of the three hospitals’ IPC programmes. These included the lack of a dedicated budget, lack of structured training for healthcare workers, lack of a professional development programme for IPC focal points, interrupted availability of IPC materials (aprons, gloves, and face masks), and lack of HAI surveillance. In light of this evidence, several recommendations ranging from low-cost to high-cost interventions were made to improve the implementation of IPC programmes in these hospitals [13].

In this study (hereafter referred to as the follow-up study), using an operational research approach, we aimed to describe the dissemination and implementation of these recommendations and measure any change in the IPC performance in the same three tertiary hospitals. Our specific objectives were to compare the baseline study, undertaken in August 2021, with this follow-up study, undertaken in March 2023, to (1) describe the dissemination activities, recommendations, and actions taken to improve the IPC programme implementation; (2) assess and compare the IPC performance scores, overall, and stratified by IPC core components; and (3) describe the current status of the gaps in the eight core components of IPC identified in the baseline study.

## 2. Methods

### 2.1. Study Design

This was a ‘before and after’ observational study using two routine cross-sectional assessments of IPC using the WHO-IPCAF tool.

### 2.2. Study Setting

#### 2.2.1. General Setting

Sierra Leone is a coastal West African country sharing borders with Guinea and Liberia. The country is divided geopolitically into five regions and 16 districts with an estimated population of eight million [14]. The World Bank estimated the country’s life expectancy at birth to be 60 years in 2020. More than half (57%) of the deaths are attributed to communicable diseases, with 16% of the Gross Domestic Product spent on health [15]. Healthcare services are largely provided by the public sector, and the public health sector is divided into primary, secondary, and tertiary care levels with about 27 government hospitals and over 1300 peripheral health units (PHUs). Primary, secondary, and tertiary care is provided at PHUs, district hospitals, and regional/national hospitals, respectively.

#### 2.2.2. Study Sites

This was a follow-up study conducted at the same three tertiary hospitals (Connaught Hospital, Princess Christian Maternity Hospital—PCMH, and Ola During Children’s Hospital—ODCH) as the baseline study [13].

Connaught Hospital, PCMH, and ODCH are public hospitals in Freetown, the capital of Sierra Leone. These are the only tertiary hospitals in Sierra Leone. Connaught Hospital has about 300 beds in 16 wards and 25 sub-units where both in- and outpatient services for medicine and surgery are provided [16]. PCMH has a bed capacity of 140 in eight wards and provides services for obstetrics and gynaecology, while ODCH has 139 beds in eight wards and provides services for paediatrics [17,18]. All three hospitals are teaching hospitals and part of the University of Sierra Leone Teaching Hospital Complex (USLTHC), established to support postgraduate clinical training.

All three hospitals have established IPC programmes with IPC committees and designated IPC focal points, whose key responsibilities are coordinating and implementing IPC activities at the hospital level.

### 2.3. Study Period

The baseline study was undertaken in August 2021, and the follow-up study was undertaken in March 2023.

### 2.4. Data Collection and Analysis

#### 2.4.1. Dissemination Activities, Recommendations, and Actions Taken

A training on the development and use of communication tools was conducted during a Structured Operational Research Training IniTiative (SORT IT) course module in May 2022, after publication of the baseline study [19]. These tools were used to disseminate the baseline research findings and recommendations to the identified decision-makers, influencers, and other relevant stakeholders. The dissemination details such as (1) mode of delivery (material used for dissemination), (2) to whom the findings and recommendations were disseminated, (3) where the dissemination was performed, and (4) when the dissemination was performed were extracted from the UN Special Programme for Research and Training in Tropical Disease (TDR) monitoring and evaluation (M&E) routinely collected data. We also collected usage statistics from social media platforms as well as citations from the published paper.

The list of recommendations made was prepared by reviewing the baseline study manuscript and dissemination materials. A record of actions taken was obtained from the original principal investigator, and the TDR M&E routinely collected data to supplement this information.

#### 2.4.2. Measurement of IPC Compliance Using the IPCAF Tool

The IPCAF tool is a self-assessment tool divided into eight sections with 81 indicators reflecting the eight WHO ‘Core Components of Infection Prevention and Control Programmes’. These are

Core component (CC) 1: IPC programme;CC2: IPC guidelines;CC3: IPC education and training;CC4: Healthcare-associated infection surveillance;CC5: Multimodal strategies for implementation of IPC interventions;CC6: Monitoring/audit of IPC practices and feedback;CC7: Workload, staffing, and bed occupancy;CC8: Built environments, materials, and equipment for IPC.

The IPCAF Tool has a total score of 100 points assigned for each CC, and thus the highest possible overall IPCAF score (IPC compliance score) for all eight components is 800. The tool has a grading system for the overall compliance score denoting a range from inadequate to advanced IPC compliance (Table 1). For this follow-up study, we applied the same approach at the core component level to grade the individual core components based on the obtained score using the following scale: (i) inadequate (0–25); (ii) basic (25.1–50); (iii) intermediate (50.1–75); and (iv) advanced (75.1–100). In presenting these categories, we developed a colour coding scheme from red denoting inadequate through green denoting advanced.

This follow-up study used routinely collected data by the IPC focal points within each hospital using the WHO IPCAF tool in consultation with relevant key stakeholders. The completed information was cross-validated by the principal investigator through a review of documents and direct observation where necessary.

The data collected were entered into a Microsoft Excel spreadsheet by the lead researcher. To ensure data accuracy, the principal investigator undertook a review of the datasets to detect any missing or incomplete data. Where possible, scores were validated by reviewing supporting documentation (for example, the training logs). The overall and core-component scores were calculated for each hospital. A descriptive analysis of each core component of the IPC programmes in all three hospitals was performed, followed by a comparative analysis with the findings from the baseline operational research. In our results, we reported the actual scores and calculated the absolute percentage change in IPC compliance.

## 3. Results

### 3.1. Dissemination Activities and Recommendations

The research team of the baseline study [13] used several dissemination methods and materials to ensure adequate uptake of research findings and recommendations. These included a publication, an elevator pitch, a plain English summary, and a Microsoft PowerPoint 2013 presentation, the latter being the most common method used. The baseline research findings and recommendations were presented to a wide range of audiences that included researchers, academicians, policymakers, and healthcare workers at different times and places (Table 2).

Out of the eight recommendations divided into low, medium, and high-cost from the baseline study, two had been fully implemented and four had been partially implemented (Table 3) [13].

### 3.2. Overall and Individual Core Component Compliance

The overall IPC compliance scores for Connaught Hospital, ODCH, and PCMH in 2023 were 482.5, 458.5, and 511, respectively, with PCMH, which had the lowest score in 2021, having the highest score in 2023. All three hospitals fell within the range of the ‘Intermediate’ level (400–600) according to the WHO IPCAF grading. There was an improvement in IPC compliance from 2021 to 2023 in all three hospitals as shown by the increased absolute percentage scores, with PCMH showing the highest improvement of 26.9% (Table 4).

There were increased compliance scores across all eight core components in the three hospitals’ IPC programmes from 2021 to 2023, with the majority of the scores ranging from ‘Basic’ to ‘Intermediate’. A two-level improvement was seen in the ‘IPC guidelines’ (CC2) and ‘multimodal strategies’ (CC5) compliance from ‘Basic’ to ‘Advanced’ for the three hospitals, making them the core components with the highest level of compliance. Improved compliance from ‘Inadequate’ to ‘Basic’ was also seen in the ‘Monitoring/audit of IPC practice’ (CC6) for the three hospitals’ IPC programmes. Only PCMH recorded an improvement in ‘HAI surveillance’ (CC4) and ‘IPC education and training’ (CC3) compliance (Table 5).

### 3.3. Baseline Gaps Status in 2023

In the baseline study, 22 gaps were documented in the different components of the IPC programmes in the three hospitals. Of these, five had been addressed whilst the others still existed in 2023. Improvements were seen in ‘IPC guidelines’ (CC2) and ‘multimodal strategies’(CC5) as more than half of the gaps no longer existed. The core components with little or no improvement were the ‘IPC programs’ (CC1), ‘workload, staffing and bed occupancy’ (CC7), and ‘built environment, materials, and equipment’ (CC8) (Table 6).

## 4. Discussion

This is the first follow-up study assessing the change in IPC programme implementation at the tertiary hospitals of Sierra Leone following a baseline operational research. We showed that several communication materials were used to disseminate the baseline study findings [13], which enabled the uptake of recommended actions. Additionally, there was an improvement in the overall IPC compliance score for each hospital as they all moved from ‘Basic’ to ‘Intermediate’ levels according to the IPCAF grading system.

Our study affirms the value of using the standardised IPCAF tool at regular intervals to assess IPC performance and inform change. However, using the tool to generate findings alone is not enough to foster change without dissemination of the recommendations. We recommend implementing a comprehensive communication of the findings using the tools we described here, evidence briefs, and presentations to enhance the awareness and take-up of the recommendations. Other factors that enabled this positive change included the political will of the senior leadership in the Ministry of Health and Sanitation and their involvement throughout the research cycle. Additionally, involvement of the principal investigators in both operational research studies improved our ability to describe the actions taken between the studies and make an association with the subsequent improvement in scores in some core components. We accept that direct attribution is not possible.

In these three hospitals, we observed that all the low-cost recommendations were implemented. The likely reason is that they can be undertaken within existing resources, infrastructure, and training programmes. Recommendations that were considered high-cost remained as gaps. Key among these was the need for a dedicated budget for IPC implementation at every hospital. Additionally, action is required at the national level to improve workload; staffing and bed occupancy (CC7); and environments, materials, and equipment for IPC (CC8) at these three hospitals.

Our second objective was to measure the change in compliance between the two study periods. There was a noticeable improvement in all three hospitals’ IPC programmes, with at least a 15% absolute increase in IPC performance. This implies that many of the core components of these hospital IPC programmes were implemented appropriately. We believe that the baseline study recommendations and their effective dissemination contributed to the actions taken by the different hospitals’ IPC teams, leading to an improvement in the follow-up study.

For the different core components, there was a two-level (from ‘Basic’ to ‘Advanced’) improvement in guidelines and multimodal strategy, while the least improvement was seen with HAI surveillance. The marked improvement seen in the IPC guideline was mainly due to updating and disseminating the national IPC guideline, which was being revised during the baseline study. The improvement seen in multimodal strategies can be associated with the mentorship and trainings on the multimodal strategy conducted by the lead PI of the baseline study as he noticed that it was challenging for IPC focal points to apply the concept of the multimodal strategy. The PIs of the baseline and follow-up operational research studies will continue to provide technical and operational support to the national IPC unit and hospital IPC teams to ensure the full implementation of operational research recommendations.

The recently published WHO global report on IPC implementation at the hospital level documented an improvement in IPC programmes across all six WHO regions (8). This report further highlighted those hospitals in low-income settings, such as Sierra Leone, that are yet to achieve the WHO-recommended ‘Advanced’ level. This is in keeping with our follow-up study findings as none of the three hospitals scored ‘Advanced level’.

There are still some critical gaps that must be addressed in all three hospital IPC programmes to reach the ‘Advanced’ level. These gaps include the lack of a dedicated budget, lack of regular IPC training for healthcare workers, and lack of routine HAI surveillance. Our findings are similar to a study conducted in Bangladesh where only 30% of the hospitals included in the survey conducted regular IPC training for staff, and none of the 11 hospitals had an HAI surveillance system in place [11]. We believe that the gaps that are still present needed more time and financial resources for them to be addressed. A longer period of follow-up might be advantageous to fully see the positive impact of operational research.

### Recommendations for Policy and Practice

For these hospitals to attain the ‘Advanced’ level according to the IPCAF grading system, we therefore recommend the following:

First, there should be continued dissemination of both the baseline and follow-up operational research findings to improve awareness. Additionally, to ensure appropriate uptake of the recommendations, there should be constant engagement with the hospital management and other IPC stakeholders. Second, there should be a dedicated budget in each of these hospitals for improvement in the implementation of IPC core components. Third, the hospital management should prioritise HAI surveillance activities and better access to international funding to improve microbiology capacities: funding is partially supported in a current Fleming Fund grant. Finally, some actions can only be taken at the national level including the improvement of workload; staffing and bed occupancy (CC7); and environments, materials, and equipment for IPC (CC8).

Our study has several strengths. First, data collection was carried out by IPC focal persons and validated by the principal investigator, who has good knowledge of IPC. Second, we used a structured and validated data collection proforma, the WHO IPCAF tool, which facilitated an appropriate comparison with the baseline study in which the same tool was used. Third, the recommendations from the baseline study were clearly stated as low cost through to high cost, supporting the investigators in assessing the status of recommendations and effective actions taken. Fourth, we adhered to ‘STROBE’ (Strengthening the Reporting of Observational Studies in Epidemiology) guidelines for data collection and the reporting of study findings [20].

There are limitations to our study. While this survey covers all three tertiary hospitals in Sierra Leone, we recognise that this is a small sample size compared to other national surveys. A qualitative approach would have added more value to our findings in identifying the root causes of poor performance in IPC implementation. The IPCAF tool is a self-assessment tool and might introduce some biases as responses given by facilities cannot be easily verified. Whilst we can mitigate such limitations by a cross-validation process, we cannot eliminate them.

## 5. Conclusions

There was an improvement in the implementation of the IPC’s eight core components at Connaught Hospital, PCMH, and ODCH as they moved from the ‘Basic’ to ‘Intermediate’ level according to the IPCAF grading from the years of 2021 to 2023. This improvement can be attributed to several factors including the recommendations made from the baseline operational research study, the actions taken, the dissemination of research findings, and the continued technical support to the national IPC unit and hospital IPC programme by both operational researchers. Finally, we have demonstrated the importance of operational research to monitor and improve programme implementation at the hospital level, which can also inform recommendations for actions at the national level.

## Figures and Tables

**Table 1 tropicalmed-08-00378-t001:** The WHO IPCAF tool grading and interpretations.

Score	Grading	Interpretation
0–200	Inadequate	Implementation of IPC core components is deficient. Significant improvement is required
201–400	Basic	Some aspects of the IPC core components are in place but not sufficiently implemented. Further improvement is required
401–600	Intermediate	Most aspects of the IPC core components are appropriately implemented. The facility should continue to improve the scope and quality of implementation and focus on the development of long-term plans to sustain and further promote the existing IPC programme activities
601–800	Advanced	IPC core components are fully implemented according to the WHO recommendations and appropriate to the needs of the facility

IPC—Infection Prevention and Control; WHO—World Health Organisation; IPCAF—Infection Prevention and Control Assessment Framework.

**Table 2 tropicalmed-08-00378-t002:** Dissemination activities of the 2021 baseline study at the three tertiary health facilities in Sierra Leone by Kamara et al., 2022 [13].

Mode of Delivery	To Whom	Where	When
PowerPoint presentation before publication	National IPC Coordinator, national IPC officers (presented to 10 people)	Public Health Emergency Operation Centre	November 2021
Published research article	Researchers/Academicians (cited by 3, viewed by 1646 on the 24 June 2023)	IJERPH	April 2022
Distribution of published article	Healthcare Workers, Young Professionals, Researchers, and professional colleagues (distributed to 2500 people)	WhatsApp groups, LinkedIn, Facebook	April 2022
10 min technical PowerPoint Presentation Elevator Pitch during coffee breaks	Researchers and AMR advocates (presented to 40 people)	International Conference: Solutions to AMR Social Sciences in Copenhagen, Denmark	October 2022
10 min technical PowerPoint presentation	National IPC team, management of hospitals and partners (presented to 50 people)	National SORT IT Dissemination Meeting	November 2022
Distribution of published article and 10 min technical PowerPoint presentation	Hospital Medical Superintendents and Deputy Chief Medical Officer-Clinical (distributed to 4 people)	Email exchange and WhatsApp	January 2023
10 min technical PowerPoint presentation by IPC unit	National IPC Officers and hospital focal points (presented to 10 people)	National IPC Unit Office	January 2023
Distribution of published article	Hospitals’ IPC focal points WHO AFRO IPC Team (distributed to 3 people)	WhatsAppEmail exchange	January 2023
Policy Brief	Researchers, AMR advocates, and community	WHO Sierra Leone website Breakthrough Action Website	March 2023
Policy Brief	SHEA Board of Trustees and International Ambassadors (distributed to 17 people)	SHEA Spring Conference	April 2023
Research article and policy brief	AMR short course participants and instructors (distributed to 21 people)	Institute of Tropical Medicine, Antwerp, Belgium	May 2023

IPC: Infection Prevention and Control; AMR: Antimicrobial Resistance; SORT IT: Structured Operational Research and Training Initiative; WHO AFRO: World Health Organization Africa Region; IJERPH: International Journal of Environmental Research and Public Health, SHEA: Society for Healthcare Epidemiology of America; ITM: Institute of Tropical Medicine, Antwerp, Belgium.

**Table 3 tropicalmed-08-00378-t003:** List of recommendations from the baseline study conducted in 2021 for improving IPC performance in three tertiary health facilities in Sierra Leone and status of actions as of April 2023 [13].

Cost of Implementation	* Recommendation	Action Status	Details of Action
Low	New employee orientation and training for all healthcare workers and administrative staff	Fully Implemented	Orientation training conducted for new staff.
	Continuous professional development programme for hospital IPC focal persons to improve their knowledge and understanding of IPC	Partially Implemented	Monitoring and evaluation trainings conducted for all the hospital IPC focal points.
Medium	Development of a national Healthcare-Associated Infection (HAI) surveillance strategy	Partially Implemented	Funding was secured to develop a national HAI surveillance strategy, and planning initiated
	Conduct regular HAI surveillance	Partially Implemented	Only surgical site infection surveillance conducted at PCMH.
	Quarterly implementation of the WHO IPCAF tool at healthcare facilities to monitor the implementation of IPC Programs	Fully Implemented	Routine quarterly assessments are conducted in the three tertiary hospitals using an adapted WHO IPCAF tool
	Uninterrupted supply of IPC materials, such as examination gloves, face masks, aprons, and other IPC materials, to protect healthcare workers	Not Implemented	Budgetary constraints
High	Ministry of Health and Sanitation and its implementing partners to provide technical and financial support (especially a dedicated budget for IPC) to the national and hospital IPC programmes for the implementation of the IPC activities at healthcare facilities to reduce the burden of HAI and AMR.	Partially Implemented	Technical and financial support provided to the national IPC unit and technical support to hospital IPC programmes. However, there is no dedicated budget at the facility level.

* Source [13]. Fully implemented—all actions taken and no further intervention required; partially implemented—actions taken but needs further intervention; not implemented—no action taken, IPC—Infection Prevention and Control; IPCAF—Infection Prevention and Control Assessment Framework; low cost—activities undertaken within the existing resources, infrastructure, and training programmes; medium cost—activities requiring a moderate increase in existing resources, infrastructure, and training programmes; high cost—the creation of a dedicated budget, increased funding, and improvement to existing workforce and infrastructure.

**Table 4 tropicalmed-08-00378-t004:** Comparison of 2021 and 2023 IPC compliance scores as measured by the IPCAF tool at three tertiary hospitals in Freetown, Sierra Leone.

Facilities	2021 *		2023		Absolute Percentage Change
	IPCAFScore N = 800 (%)	Interpretation	IPCAF Score N = 800 (%)	Interpretation	
Connaught Hospital	333.5 (41.7)	Basic	482.5 (60.3)	Intermediate	+18.7
Ola During Children Hospital (ODCH)	323.5 (40.7)	Basic	458.5 (57.3)	Intermediate	+16.9
Princess Christian Maternity Hospital (PCMH)	296.0 (37.0)	Basic	511.0 (63.9)	Intermediate	+26.9

* Source (13). IPC—Infection Prevention Control; IPCAF—Infection Prevention and Control Assessment Framework at the facility level. Maximum IPCAF score was 800: 0–200 Inadequate; 201–400 Basic; 401–600 Intermediate; and 601–800 Advanced.

**Table 5 tropicalmed-08-00378-t005:** Comparison of 2021 and 2023 compliance for each core component of the IPC programs by the IPCAF tool at three tertiary hospitals in Freetown, Sierra Leone.

Core Components	Facility Name and IPCAF Interpretation					
	Connaught		ODCH		PCMH	
	2021	2023	2021	2023	2021	2023
CC1: IPC programme	Intermediate	Intermediate	Intermediate	Intermediate	Intermediate	Intermediate
CC2: IPC guideline	Basic	Advanced	Basic	Advanced	Basic	Advanced
CC3: IPC education and training	Basic	Basic	Basic	Basic	Basic	Intermediate
CC4: HAI surveillance	Inadequate	Inadequate	Inadequate	Inadequate	Inadequate	Intermediate
CC5: Multimodal strategies	Basic	Advanced	Basic	Advanced	Basic	Advanced
CC6: Monitoring/audit of IPC practice	Inadequate	Basic	Inadequate	Basic	Inadequate	Basic
CC7: Workload, staffing, and bed occupancy	Basic	Basic	Basic	Basic	Basic	Basic
CC8: Built environment, materials, and equipment	Intermediate	Intermediate	Basic	Intermediate	Basic	Intermediate
Overall score	Basic	Intermediate	Basic	Intermediate	Basic	Intermediate

IPC—Infection Prevention Control; IPCAF—Infection Prevention and Control Assessment Framework at the hospital level. Maximum IPCAF score was 800: 0–200 Inadequate; 201–400 Basic; 401–600 Intermediate; and 601–800 Advanced. ODCH–Ola during Children’s Hospital; PCMC–Princess Christian. Legend: 
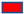
 Inadequate, 
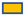
 Basic, 
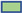
 Intermediate, 
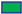
 Advanced.

**Table 6 tropicalmed-08-00378-t006:** The 2023 status of the gaps identified in the different components of the IPC programmes at the three tertiary hospitals in Freetown, Sierra Leone in 2021.

Core Components	Components of Hospitals IPC Programmes	
	Gaps in 2021 *	Status in 2023
IPC program	No dedicated budget for the IPC programme	The gap still existed
IPC guideline	No written guidelines for Outbreak management and preparedness. Prevention of the different types of HAI	Available guidelines for the prevention of the different types of HAI were in the updated national IPC guidelines. However, there were no written guidelines for outbreak management and preparedness
IPC education and training	No regular IPC training was conducted for healthcare workers and administrative staff IPC training was not integrated into clinical practice, as well as the training of specialists No IPC training for patients or family members to minimise HAI No certified continuous professional development courses for IPC focal persons	Three out of the four gaps existed as only health education had been conducted for patients and family members to minimise HAI
HAI surveillance	No information technology support to conduct surveillance activities No HAI surveillance was being conducted by hospitals except for PCMH conducting SSI surveillance No analysis of antimicrobial drug resistance data, due to a lack of microbiology capacity	Two out of the three gaps still existed as there was available information technology support to conduct surveillance activities in all the hospitals
Multimodal strategies	Safety climate and culture change were not included in the multimodal strategy A multidisciplinary team was not used to implement the multimodal strategies	A multidisciplinary team was used to implement the multimodal strategy. However, there was still a need for safety climate and culture change to be included in the multimodal strategy
Monitoring/audit of IPC practice	No defined monitoring plan with clear goals, targets, and activities No hospitals monitored: Intravascular catheter insertion and/or care; wound dressing drainage; and consumption of alcohol-based hand rub	Only one (PCMH) out of the three hospitals had a defined monitoring plan with clear goals, targets, and activities. Intravascular catheter insertion and/or care; wound dressing drainage; and consumption of alcohol-based hand rub were not monitored in all three hospitals
Workload, staffing and bed occupancy	Staffing levels were not assessed according to patient workload and there was no agreed healthcare-worker-to-patient ratio across the hospitals No system in place to assess and respond when bed capacity was exceeded Inadequate bed spacing in certain departments across all the hospitals	All the gaps still existed
Built environment, materials and equipment	No reliable safe drinking water always available for staff, patients, and family members and in all locations No single-patient rooms for grouping patients with similar pathogens The constructed burning pit/waste dump in the hospitals had insufficient dimensions Non-functional incinerators in the hospitals Disposable items, such as examination gloves, facemasks, and aprons, were not continuously available	Only Connaught Hospital had a functional incinerator

* Source [13]. IPC—Infection Prevention Control; IPCAF—Infection Prevention and Control Assessment Framework at the hospital level. Maximum IPCAF score was 800: 0–200 Inadequate; 201–400 Basic; 401–600 Intermediate; and 601–800 Advanced. In this study, a gap was assigned to those scoring below 25 in any core component.

## Data Availability

The dataset used in this paper has been deposited at https://figshare.com/account/items/23511639/edit#:~:text=Close-,10.6084/m9.figshare.23511639,-Copy%20DOI (accessed on 4 February 2021) and is available under a CC BY 4.0 license.

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
