# Peer review of "Improvement in Infection Prevention and Control Compliance at the Three Tertiary Hospitals of Sierra Leone following an Operational Research Study"

_tropicalmed, 2023, doi:10.3390/tropicalmed8070378_

Round 1
Reviewer 1 Report
The manuscript titled "Improvement in Infection Prevention and Control Compliance at the Three Tertiary Hospitals of Sierra Leone Following an Operational Research Study" deals with an interesting topic. It is a follow-up study and the authors aim to compare a baseline study, undertaken in August 2021, with this study, undertaken in March 2023 to: describe the dissemination activities, recommendations and actions taken to improve the IPC programme implementation; assess and compare the IPC performance scores; and describe the current status of the gaps in the eight core components of IPC identified in the baseline study. The study presents some limitations which have been clearly indicated in lines 347-353.
The paper is well-written but it needs some changes and clarifications:
- Table 1: The interpretation of the grading "Advanced" of WHO IPCAF tool is reported as "IPC sufficiently implemented. Further improvement is required" but, according to the original tool, it should be "The IPC core components are fully implemented according to the WHO recommendations and appropriate to the needs of the facility". Please, resolve this inconsistence.
- Lines 183-185: it is reported that "In presenting these categories, we developed a colour coding scheme from red denoting inadequate through green denoting advanced (Table 4)." but the colour coding scheme is present in Table 5 and not 4. Please, resolve this inconsistence.
- Table 3: The action status of the recommendation "Quarterly implementation of the WHO IPCAF tool at healthcare facilities to monitor the implementation of IPC Programs" is reported as "Implemented" instead of "Fully implemented", as reported in line 214. Please, resolve this inconsistence.
Reviewer 2 Report
Well conducted study, with a sound methodology and clearly presented results. There is important improvement in several CCs across all 3 included hospitals, which shows that the baseline study had a positive effect.
I have to comments/suggestions to address in the discussion
- whether changes and measures applied during the COVID19 pandemic might have had an effect on the findings of this follow-up study.
- whether there are other follow-up studies from other countries in the literature that have shown any changes in their IPCAF findings.
